# Pandemics and Burden of Stroke and Epilepsy in Sub-Saharan Africa: Experience from a Longstanding Health Programme

**DOI:** 10.3390/ijerph18052766

**Published:** 2021-03-09

**Authors:** Massimo Leone, Fausto Ciccacci, Stefano Orlando, Sandro Petrolati, Giovanni Guidotti, Noorjehan Abdul Majid, Victor Tamba Tolno, JeanBaptiste Sagno, Darlington Thole, Fabio Massimo Corsi, Michelangelo Bartolo, Maria Cristina Marazzi

**Affiliations:** 1The Foundation of the Carlo Besta IRCCS Neurologic Institute, 20133 Milan, Italy; 2UniCamillus Saint Camillus International, University of Health Sciences, 00100 Rome, Italy; fausto.ciccacci@gmail.com; 3University of Rome Tor Vergata, 00100 Rome, Italy; stefano.orlando@dreameurope.org; 4San Camillo Hospital Department of Cardioscience, 00100 Rome, Italy; sandropetrolati@gmail.com; 5Azienda Sanitaria Locale (ASL) Roma 1 Regione Lazio, 00100 Rome, Italy; giovanni.guidotti@dreamsantegidio.net; 6Community of S. Egidio DREAM Program, Maputo 1102, Mozambique; noorjehanmagid@dream.org.mz; 7Community of S. Egidio DREAM Program, Blantyre 312224, Malawi; tolnovictortamba@gmail.com (V.T.T.); sagnojb@gmail.com (J.S.); 8Community of S. Egidio DREAM Program, Balaka 302100, Malawi; darlthole@gmail.com; 9Salvator Mundi International Hospital Neurology, 00100 Rome, Italy; fmcorsi@gmail.com; 10Telemedicine Department San Giovanni Addolorata Hospital, 00100 Rome, Italy; mbartolo@hsangiovanni.roma.it; 11Libera Università Maria SS. Assunta, 00100 Rome, Italy; mcmarazzi@gmail.com

**Keywords:** sub-Saharan Africa, stroke, epilepsy, HIV/AIDS, COVID-19, medical education, retention, pandemic, care disruption, DREAM programme

## Abstract

Eighty percent of people with stroke live in low- to middle-income nations, particularly in sub-Saharan Africa (SSA) where stroke has increased by more than 100% in the last decades. More than one-third of all epilepsy−related deaths occur in SSA. HIV infection is a risk factor for neurological disorders, including stroke and epilepsy. The vast majority of the 38 million people living with HIV/AIDS are in SSA, and the burden of neurological disorders in SSA parallels that of HIV/AIDS. Local healthcare systems are weak. Many standalone HIV health centres have become a platform with combined treatment for both HIV and noncommunicable diseases (NCDs), as advised by the United Nations. The COVID-19 pandemic is overwhelming the fragile health systems in SSA, and it is feared it will provoke an upsurge of excess deaths due to the disruption of care for chronic diseases such as HIV, TB, hypertension, diabetes, and cerebrovascular disorders. Disease Relief through Excellent and Advanced Means (DREAM) is a health programme active since 2002 to prevent and treat HIV/AIDS and related disorders in 10 SSA countries. DREAM is scaling up management of NCDs, including neurologic disorders such as stroke and epilepsy. We described challenges and solutions to address disruption and excess deaths from these diseases during the ongoing COVID-19 pandemic.

## 1. The Epidemiologic Transition Is Changing the Health Scenario in Sub-Saharan Africa

Sub-Saharan Africa (SSA) demography has deeply changed in the last decades. Interactions among environmental determinants also are incriminated in changes of disease burden. The SSA population almost doubled in the last 20 years, from 660 million to 1.1 billion people [1]. Life expectancy is 61 years (80.6 in high income countries—HIC, as in Europe in 1950 [2,3]), 10 years more than in 2000 [2] (Figure 1).

This deep demographic change has increased the burden of noncommunicable diseases (NCDs) that today are responsible for more than 40% of the total disease burden in SSA [4,5]. The rise of stroke and epilepsy is alarming: from 1990 to 2017 disability-adjusted life years (DALYs) increase for stroke was +37.7% and epilepsy 68.8% [4]. Stroke and epilepsy can be largely prevented and treated in SSA and deserve more attention (Figure 2).

## 2. Stroke in Sub-Saharan Africa

Stroke has become the second leading cause of death worldwide and ranks as the biggest (67.3%) contributor to the global burden of neurological disorders [6]. In the last 20 years, stroke greatly increased in low- to middle-income countries (LMIC); 80% of people who have had a stroke live there [7]. Mortality rate (number of deaths per 100,000 general population) for stroke in SSA is up to 5 times higher compared to Western countries—caused by fast demographic and lifestyle changes and other local factors including poor access to primary care [8]. In 2016, the death rate by stroke was 37 deaths per 100,000 inhabitants in SSA, similar to malaria and TB (40 deaths per 100,000 inhabitants) [5]. In 75% of SSA countries, stroke is among the three leading causes of death [8] (Figure 3).

The main causes of stroke and related risks factors in SSA differ from Western countries [9,10], and modifiable risk factors account for more than 80% of stroke risk in SSA [7]. After hypertension, HIV is the second main risk factor [9], and HIV has been shown to be the main risk factor for stroke in patients below 45 years of age [9]. Compared to Western countries, stroke is more common in young people in SSA [10]. Severity, mortality, and occurrence of stroke are increased by poor socioeconomic conditions, particularly at younger ages [11]; extreme poverty (people living with less than 1.9 USD per day) affects more than 40% of the SSA population [12].

Tobacco use, dyslipidaemia, and atherosclerosis are observed less frequently in SSA stroke patients compared to Europe [7,9].

Although more than 50% of the population is under 20 [13], hypertension affects between 16% and 40% (median 31%) of the adult SSA population; most of them are not aware of their condition [14,15]. The same applies to diabetic patients (8% of general population) [15]. In general, more than half of the SSA population live in rural areas with poor access to health care, so that most hypertension and diabetes—major contributors to stroke—remain undisclosed and poorly treated and controlled [15].

Integrated prevention programmes in primary care settings have the potential to greatly improve the condition in SSA.

## 3. Epilepsy in Sub-Saharan Africa

Epilepsy is a chronic NCD of the brain, one of the most common neurological diseases [16]: in 2016 there were 45.9 million patients with active epilepsy worldwide [17]. Eighty percent of epileptic patients live in poor countries, from 5 to 12 million in SSA [17,18,19]. In many regions of SSA, idiopathic epilepsy represents the second most frequent neurological disorder [6,17]. The risk of premature death among people with epilepsy is 3 times higher than in the general population and even higher in SSA [20]. More than one- third of all epilepsy-related deaths occur in SSA [19]. In a global burden of disease review, epilepsy was between the 16th and 25th leading cause of death in SSA, while in HIC it was ranked between 43 and 63 [21].

The treatment gap (people with epilepsy not getting the treatment they need) exceeds 70% in most of the SSA countries [16,18], and up to 70% of these epileptic patients could live seizure-free if properly diagnosed and treated [16].

Low adherence to antiepileptic drugs (AEDs) [22] is among the causes favouring convulsive status epilepticus [23], particularly in HIV and malaria patients, which in turn increases the risk of brain damage and death [24]. Interventions to improve medication adherence in epilepsy are very limited in SSA [25]. A short course health education programme did not improve adherence in SSA, suggesting the need for long-term interventions [26].

A shortage of antiepileptic medicines in public health centres is a challenge in SSA, and in some cases lack of drugs may undermine the effectiveness of the same. Many people think epilepsy is the result of witchcraft, which leads parents of children with epilepsy to seek the services of traditional healers [16]. Those children cannot go to school because of stigma, so that the illness brings the hard consequences of stigma as illiteracy, restriction in social contacts, marriage, work, etc. [23]. Patients and their families very often suffer from stigma and discrimination too. Resources to treat an epileptic patient in Western countries and SSA differ considerably: 2051–11,354 USD [27] and <2 USD [28], respectively. The disparity underlines the need for more investments to treat epilepsy in SSA.

## 4. The Double Burden of HIV and Neurologic Disorders in Sub-Saharan Africa

The double burden of HIV and stroke and epilepsy challenges patients, health care services, and governments in SSA. It has been shown that HIV infection is a risk factor for stroke [9] and epilepsy [29] even when viral load is undetectable. Small amounts of virus remain in the nervous system of HIV patients permanently to interfere to various extents with normal functioning of neurons, glial cells, and vessels, thus predisposing millions of HIV patients to neurologic disorders [30]. The lasting HIV pandemic will further increase the burden of stroke and epilepsy in SSA. In fact, the total number of people living with HIV (PLWH) continues to increase as the result of longer life expectancy due to more effective antiretroviral treatment regimens and the decreased occurrence of new infections [31,32]. New HIV infections decreased from 1.9 in 2015 to 1.7 million in 2019, about a 2%/year reduction rate [33]. At this rate, in 2050 new infections will still be about 1 million/year. On the other hand, the number of PLWH continued to increase from 30.7 in 2010 to 38 million in 2019, even faster than what has been observed from 2000 (24 million) to 2010: 7.3 million increase from 2010 to 2019 and 6.7 million from 2000 to 2010) [33] (Figure 4). In 2019, there were 26 million PLWH in SSA (68% of the total) [31].

To reduce the number of PLWH, renewed efforts are needed to prevent new infections. The most effective HIV prevention is antiretroviral treatment (ART) [34]; bringing ART to all patients living with HIV in SSA patients in the period 2018–2050 will cost about 373 billion USD [35]. SSA governments and development assistance for health (DAH), the two major sources covering health expenditure in SSA, could provide a total of 248.6 billion USD in that period (according to the yearly change observed in the last five years in HIV expenditure of SSA governments plus DAH, +3.3% and −1.9%, respectively—reference period 2010–2017) [36] so that 124.4 billion USD will be lacking. In addition, to effectively reduce new infections, the major share of the resources will be needed in the next few years [35].

## 5. Unifying Care of HIV, Stroke, and Epilepsy at HIV Centres: Toward Resilient Health Systems

To fight the increasing double burden of HIV and NCDs, the United Nations indicated to unify management of HIV and NCDs along with stroke and epilepsy at HIV centres [37]. There are some issues in the process of building services and providing management for stroke and epilepsy at HIV centres, among which task-shift, retention, and poverty deserve special attention.

### 5.1. Shortage of Neurologists in SSA: Education, Training, and Task-Shift

In SSA the number of medical doctors is limited, and neurologists are exceptionally rare, about 1 for every 3–5 million people [38] (Figure 5).

According to Western countries’ standard (4.75 neurologists per 100,000 inhabitants) SSA would need 49,000 additional neurologists [38]. Western countries’ governments spend not less than 160,000 euros to graduate one student [39], that is 7.8 billion Euros to graduate 49,000 students—further expenses should be added to become neurologists. This amount of money corresponds to the 2019 GDP of a SSA country such as Malawi [40]. The large-scale emigration of physicians, the *brain drain*, from SSA is another development concern [41]. In 2011, it was calculated that more than 7000 physicians from SSA appeared among the US physician work force; 1.2% (86/6888) were neurologists [41]. The length of service they provided to their home country in SSA before migration has been reducing from an average of 5.7 years in the period 1995–1999 to 1.3 years in the period 2005–2008 [41]. In a *best scenario*, it has been estimated that 30 years will not be enough to fill the gap of neurologists in SSA [42]. Given the dramatic shortage of neurologists, the vast majority of stroke and epilepsy cases are routinely managed by nonmedical health workers (middle level health workers or clinical officers). To improve care for stroke and epilepsy, basic neurology knowledge has to be delivered to this cadre of health workers.

A systematic analysis conducted on 195 countries to evaluate performance on access to and quality of healthcare, including stroke and epilepsy, showed that SSA countries scored at the bottom of the list [43] (Figure 6).

The education and training of health workers have been ranked as the top key factors necessary to increase quality of and access to integrated health services in low- and middle-income countries (LMICs) [44]. *Postcolonial* medical education represents a challenge for Western countries’ academic institutions due to the peculiar different contexts of SSA [45]. It has been reported that education strategies and curricula should not be “precooked” in Western universities; approaches to education should be context-specific and modelled according to practice context [45]. Lack of extended residencies in LMICs (impeded by HIC institutional policies), insufficient value attributed to training and mentoring LMIC health workers, and inadequate funding for LMIC institutions and collaborators compared to funding for HIC institutions are among the barriers in the task-shift process from Western specialists to local health workers [46]. For instance, no African countries benefited from the 1 billion USD invested to fight Ebola [47], leaving local health systems largely as weak as they were before the pandemic, and unprepared for a new pandemic [48].

Brief, standalone trainings are perceived to be insufficient by health workers [44]; after-training follow-ups enhance performance but need to be frequent and not far from each other [44]. Improved knowledge of the context and strengthening the relationship with local health workers enable the task-shift process [49].

### 5.2. Retention

Unifying management of HIV and NCDs in SSA is hampered by poor patient retention: only 60% of people who access ART in low- and middle-income countries are still receiving the treatment 4 years after its beginning [50,51] (Figure 7).

Poor retention favours the spread of HIV to the unaffected population; it increases resistance to ART, opportunistic infections, and admissions; and impoverishes households.

At a main tertiary hospital in Zambia, despite the availability of care, less than half of HIV-infected people with new-onset seizures were actively engaged in care [53].

In a programme to impact the burden of hypertension and diabetes in Malawi, less than half of the individuals started on antihypertensive medication were still in contact with the clinic 24 months after initiating treatment [52].

### 5.3. Poverty

Another barrier affecting chronic care of both communicable diseases and NCDs (including stroke and epilepsy) is the payment requirement to access health services [54,55]. In SSA, 41% of the population lives below the poverty threshold and cannot afford the cost of health care [12]; hence, many households encounter a catastrophic economic expenditure when facing the costs and consequences of chronic diseases (costs for medications, investigations, examinations, travel to health centres, loss of productivity, children not going to school, etc.) [54]. Three key preconditions for catastrophic payments have been identified: the availability of health services requiring payment, low capacity to pay, and the lack of prepayment or health insurance [56]. All of these are largely present in SSA [12]. After the abolition of fees, the use of public health services substantially increased among the poor population [57].

## 6. The DREAM Program Experience in Unifying Management of HIV and Neurologic Disorders at HIV Centres

Disease Relief through Excellent and Advanced Means (DREAM) is a health programme started in 2002 and now active in 10 SSA countries to provide health services to prevent and treat HIV/AIDS and other chronic diseases [58]. Since its onset, DREAM has opted to deliver all services for free in order to avoid catastrophic health expenditure [57] and to grant access to all [58] in accordance with the declaration of universal access to essential primary health care.

Over 500,000 people infected with HIV are receiving antiretroviral treatment from DREAM centres, more than 100,000 of whom are under 15 years of age [58,59]. All the 50 DREAM health centres and the 28 laboratories where viral load is routinely detected are headed by qualified local personnel. Due to the doctors shortage, care is largely provided by local nonmedical health workers (clinical officers).

### 6.1. Education and Training of Local Health Personnel

In DREAM, education and training of local health personnel has been committed from the outset to providing high quality management [59]. Context-specific education and training are delivered as a continuous process with frequent follow ups, shared field work, and practice including distancing learning, continuing education, and clinical case discussions. Teaching courses and education are free and open to non-DREAM personnel from other health facilities. DREAM doctors are also hosted in Europe to receive education and training on NCDs [59] (for further details about education and training in DREAM, see the Telemedicine section). In the last years, teaching and training courses have both been expanded to include stroke and epilepsy [58]. The long duration of the programme (beyond a self-ending project) enables task-shifting from Western specialists to local health workers [44,45]. Continuous education in neurology (as well as other specialties) is performed by volunteer European specialists spending periods (at least one month a year) at DREAM.

More than 10,000 African personnel, including clinical officers, doctors, nurses, biologists, laboratory technicians, administrators, home care assistants, and computer experts, have been trained by DREAM in over 120 local training courses [58,59,60,61] (Figure 8).

This health workforce makes it feasible to scale-up health programmes for conditions that need continuous care, such as HIV and many NCDs, including stroke and epilepsy [42,60,61,62,63]. The trained personnel routinely check blood pressure, glucose, and BMI in all DREAM patients, and thousands of patients suffering from arterial hypertension and diabetes regularly receive treatment, education on diet, and counselling to avoid smoking and sedentary life to prevent stroke and other cardiovascular diseases. Education on epilepsy and related stigma is provided, and this may improve adherence and retention [25]. Regarding cervical cancer, thousands of women have been tested at DREAM centres for early diagnosis and treatment of cervical cancers. Local and national campaigns on TV, newspapers and social media have been employed; governments, district hospitals, and health facilities have been involved in those campaigns [58].

### 6.2. DREAM Policies Designed to Retain Patients

To fight the unsatisfactory adherence and retention to chronic care [50,51], DREAM adopts strategies involving dedicated health personnel and local communities [58,59,64].

#### 6.2.1. The Expert Clients

Some HIV DREAM patients are actively engaged as expert clients to provide convincing evidence, effective assistance, and practical guidance to other HIV patients [58,59]. Two-thirds of HIV patients in SSA are women [33], as the majority of DREAM expert clients [58,59]. Their activity is particularly helpful when patients face troubles: life events such as divorce (not unusual in the fragile African families), illnesses and/or death of relatives, and food shortage, which can affect adherence to HIV treatment. When HIV patients feel better, their motivation to continue ART can be weakened, and healers may discourage patients to take ART rather than medical herbs. The expert clients provide peer counselling before, during, and after initiation of treatment. They also assist in patient care (recording of vital signs and weight) and follow-up and do home care service when patients are unable to get to the DREAM centre because of sickness, weakness from malnutrition, no money for transportation, loss of motivation to continue the treatment, etc.

#### 6.2.2. Strategies for HIV+ Young Patients

In SSA there are millions of HIV+ adolescents born from HIV+ mothers, affecting the future of the region [33]. Many are orphans and may not be well accepted at relatives’ homes. Once aware of their condition, they often develop depression and anger towards relatives and surrounding people, avoid coming to the HIV centre and even to school because they are afraid of the stigma. Retention in care of these young patients is a serious issue; they represent a particularly fragile HIV+ patient subgroup at increased risk of neurologic diseases such as epilepsy and stroke. Personalized strategies to retain HIV+ adolescents are adopted at DREAM centres, including special days dedicated to them [58] to help with school activities and to talk to each other; this helps fight isolation and stigma. Some of HIV+ adolescents give talks to schools and disclose their condition to show that a good life is possible with HIV. This strengthens their motivation to maintain good adherence so that constant undetectable viral load much reduces the risk of developing neurologic disorders.

#### 6.2.3. Fighting Malnutrition

Malnutrition is an endemic problem in SSA, and food supplementation is provided by DREAM to severely malnourished HIV patients, pregnant women, children, the elderly, and when clinical conditions are poor [58].

Telemedicine increases confidence in the health care system to favour retention (see Telemedicine section).

As the result of the abovementioned strategies and actions, only 1.3% of patients per year are lost to follow-up from DREAM centres [64], much less than what is reported in SSA [51]. Good retention and adherence to ART have led to 98% of HIV+ mothers surviving four years after delivery in DREAM [59]. Additionally, infant mortality has much decreased thanks to the sharp reduction in vertical transmission of HIV—less than 5% at 24 months in infants breastfed by HIV+ mothers [60], leading to more than 120,000 virus-free children born to infected mothers [58].

### 6.3. Telemedicine

A telemedicine system was started to support clinical activities in DREAM and to also increase the quality of health care in remote areas. DREAM now has 30 telemedicine workstations in SSA [65].

Teleconsultations consist of written requests sent by local clinical officers or doctors to specialists in Europe through a dedicated web platform [60,65]. Along with clinical history and examination, a number of laboratory data are automatically downloaded from the DREAM software (computerized clinical record of the patient) (Figure 9) such as viral load, CD4, transaminases, creatinine, hemogram, and other laboratory parameters, as well as information about past and ongoing treatments, BMI, growth in children, pregnancy, adherence to the antiretroviral treatment, and personal and social information. Images and videos can also be sent [60,65] (Figure 10).

A triage code is assigned to each request. Real time discussion of cases is also possible [60,65]. To receive advice from cardiologists and neurologists, health personnel are trained to perform and send through the telemedicine system electrocardiograms in all patients with cardiovascular risk factors [60,65]. In more than two-thirds of the requests, opinions from more than one specialist are requested and multi-specialist advice on the patient is offered [60]. Each specialty has a coordinator to facilitate the question-and-answer process, and a moderator (a doctor in Europe) leads the web-discussion. In addition to shared work on the ground, phone calls, whatsapps, and emails reinforce communication and relationship between local health care givers and specialists in Europe [60].

Good communication and relationships between local health care workers and specialists offering their advice from Western countries enable telemedicine development in low-income countries [49], and volunteer specialists are offered education in their local context [58,60,61,62,63].

From January 2019 until October 2020, 5611 teleconsultations were provided: 541 (9.6%) were neurologic and 3657 (65.2%) cardiologic teleconsultations mainly in hypertensive HIV+ patients [65].

### 6.4. Stroke Prevention at DREAM

Arterial blood pressure (BP), blood glucose, hemogram, creatinine, urea, urinalysis, and body mass index (BMI) are regularly checked in all patients. All patients receive counselling for lifestyle change. In patients harbouring cardiovascular risk factors such as hypertension, diabetes, kidney failure, etc., electrocardiogram (ECG) is regularly performed, and ECGs are evaluated by remote cardiologists employing the dedicated telemedicine platform. Controls and visits are tailored according to risk factors and clinical condition, including BP, BMI, viral load, adherence, blood glucose, cholesterol, smoking and drinking habits, and social condition.

## 7. Disruption of Care under COVID-19 Pandemic in Sub-Saharan Africa Affects Stroke and Epilepsy

Pandemics have devastating effects on the fragile health systems in SSA. During Ebola, health systems suffered relevant care disruption—more than 90% in some areas [66], provoking an upsurge of excess (non-Ebola) deaths among chronically ill patients such as those with HIV, TB [67], cerebrovascular disease, and cancer [66]. The COVID-19 pandemic in Africa is spreading at a lower speed compared to other geographic areas in the world [68]. COVID-19 mortality in Africa is 2.4%, similar to the 2.3% in Europe [68], but this similarity may hide differences. Mortality due to COVID-19 clearly increases with age, from 0.6% under 59 to 16.6% over 80 [69]. People 60 years of age and older represent 25.3% and 4.7% of the European and SSA general population, respectively [1]. Considering the lower proportion of people over 60, COVID-19 mortality in SSA was 5.6 times higher than Europe [1,68]. Mortality among COVID-19 infected people in SSA is probably the consequence of a number of factors, including the high burden of infectious and NCDs, coupled with weak health systems offering a platform for the epidemic to hit the SSA population hard.

Recent reports confirm that the COVID-19 pandemic is disrupting chronic care in SSA. In a WHO survey, more than one-third of African nations declared not having a budget against NCDs, and 71% of African countries reported disruptions of services for chronic conditions, such as rehabilitation [70]. In Western countries, stroke ranks among the most frequent causes of non-COVID excess deaths [71,72], paralleled by a decrease of 39% in the number of patients asking for assistance at emergency stroke departments for acute stroke during COVID-19 [73]. In SSA countries such as Burkina Faso, many of the COVID-19 patients who died did not seek care at all, and hypertension and diabetes, main risk factors for stroke, were often reported as underlying conditions [74].

The lockdowns and border closures imposed to stop COVID-19 spread in Africa are impacting on the medicines supply chain, potentially leading to increases in ART cost, 10% to 25% higher than normal prices [75], and to supply issues including substandard and counterfeit medicines [76]. A six-month complete disruption in HIV treatment could lead to more than 500,000 additional deaths from AIDS-related illnesses [75]. Similarly, if services to prevent mother-to-child transmission of HIV were disrupted for six months, new HIV infections among children would be 162% in Malawi, 139% in Uganda, 106% in Zimbabwe, and 83% in Mozambique [77]. A WHO study predicted that if the supply chains for drugs and insecticide against malaria were disrupted, the malarial death toll in SSA would double in 2020—769,000 deaths [78]. Antihypertensive drugs play a key role in the prevention of stroke, particularly in SSA [14]. A quality assessment of five commonly used antihypertensive drugs in 10 sub-Saharan African countries revealed that nearly one-quarter of the available generic (70% of the total) antihypertensive drugs were found to be of poor quality, with potential deleterious consequences [79]. Potential negative effects of care disruption due to COVID-19 in resource-poor countries have been reported for epilepsy [80]. Studies conducted in Guinea Bissau and Nigeria showed that 63–88% of epileptic patients at different sites had recurrence or increased frequency of seizures, and two people died, possibly as a result of seizures, and this was largely attributable to the discovery of falsified phenobarbital tablets [81]. Counterfeit AEDs are detrimental because sudden loss of seizure control and consequent unexpected recurrence of seizures can be life-threatening; withdrawal symptoms can be severe; and it can lead to a loss of confidence in the health care system, reducing access to treatment [80]. An increase in falsified drugs on the black market during COVID-19 is feared to affect both stroke and epilepsy in SSA.

## 8. DREAM and COVID-19: The Case of Malawi

The uninterrupted activity of DREAM favours building a resilient health care system in SSA, scaling-up of best practices to manage chronic diseases, and promoting networking among local personnel. Resilient health systems have the potential to limit the catastrophic effects provoked by pandemics. During Ebola, DREAM adopted in Guinea a number of activities to prevent care disruption, including medical triage by telephone, individualised counselling, differential programmes according to clinical and social conditions, home-based care for patients with advanced HIV, longer refills and/or home delivery of chronic medication for stable patients, focused training to health workers, enhanced link with local communities, stigma reduction messages, special attention to prevent stock-outs, and accelerated integration of communicable diseases (CDs) and NCDs diseases [82]. The combination of these actions with the preexisting trust between patients and DREAM personnel contributed to prevent care disruption and avoided excess deaths among DREAM patients [82]. A similar programme was started since COVID-19 appearance at DREAM centres in SSA, including Malawi [58].

### The Case of Malawi

Malawi is the third to last poorest country; 83% of the population live in rural areas. In 2017, the government health spending per person was 10.81 USD (2100 USD in Italy; 5400 USD in USA) [36]. In 2005, 11.7–17.1% of the population was HIV+ but only 3.1% had access to ART [83]. DREAM has introduced ART in the country since 2005 as the gold standard treatment against HIV, along with viral load monitoring, to check adherence and resistance [58]. In 2014, this approach—the test-and-treat as in Western countries—was introduced as the official country guideline [84], and nowadays 79% of the HIV+ patients receive ART [85]. These patients are at increased risk of developing stroke [9] and epilepsy [29]. In Malawi, stroke is the fifth leading cause of death [8], and HIV emerged as the most important risk factor for stroke in patients below 45 years of age [9]. The prevalence of epilepsy in the country is very high, around 2.8%, and access to anti-epileptic treatment and care is limited [86].

In Malawi, DREAM follows 17,280 patients (88% HIV+) in 15 health centres, 6% of the patients are hypertensive, and 1.1% are diabetics (unpublished observation). A neurology programme to prevent stroke and manage epilepsy is active and receives support by European neurologist volunteers [58,60,61,62]; the cooperation between neurologists, cardiologists, and local health workers at DREAM is the core of the stroke prevention programme. In 2019, a doctor from DREAM-Blantyre spent a training period on epilepsy and general neurology at the Foundation of the Carlo Besta IRCCS Neurologic Institute Milan and is sharing his knowledge in neurology at the country level [87]. In the last 4 years, 1132 teleconsultations have been sent from Malawi DREAM centres to European specialists, 19% to neurologists [65].

The Italian Society of Neurology has become partner of the DREAM epilepsy education and training programme in SSA [62] and recently donated a video EEG [88]. The main aim of this partnership is to enable the long-term process of task shifting basic neurology to local health workers [62].

As in many SSA countries, Malawi is facing the COVID-19 emergency. Difficulties in finding individual protection devices to prevent disease spread and a lack of breathing aid devices as well as emergency beds in case of COVID-19 disease are alarm bells. Many health services across the country closed or largely reduced their activities. Since March 2020, Malawi DREAM have introduced strategies to continue care delivery to thousands of chronically ill patients, adopted strict rules to prevent COVID-19 infections among patients and personnel, and initiated specific plans to enhance patients’ retention, replicating what DREAM performed during Ebola [82]. Patients harbouring cardiovascular risk factors such as hypertension, diabetes, arrhythmias, obesity, etc. receive personalized appointments for check-ups and visits according to their condition. Monitoring of epileptic patients is tailored to their preexisting adherence and responsiveness to antiepileptic drugs. Age, general and social condition, and distance from the health centre are also considered. From March to November 2020, the percentage of DREAM patients lost-to-follow-up during and before COVID-19 was 1.04% (unpublished observation) and 1.3%, respectively [64]. DREAM laboratories are supporting the diagnosis of COVID-19 (PCR diagnostic test), and more than 20% of COVID-19 genome research tests in the country have been processed at DREAM [89]. Seventy-four teleconsultations, mainly for epilepsy, have been requested to assist DREAM patients during the COVID-19 pandemic [65]; 238 additional teleconsultations to cardiologists and internal medicine specialists have been requested to prevent stroke [65]. Due to the COVID-19 travel limitations, four teaching courses on epilepsy have been delivered remotely.

DREAM continues to pay special attention to the drug supply in order to prevent stock interruption and to provide high quality drugs to avoid the feared effects of fake drugs [76].

Taken together, these activities can contribute to prevent excess deaths during the COVID-19 pandemic.

## 9. Conclusions

The SSA population is changing fast, and changes will continue in the next decades as well as local disease patterns. The SSA population and its related life expectancy is growing faster than the rest of the world. This epidemiologic transition coupled with the lasting HIV pandemic causes the worrying increase of the double burden of HIV and NCDs, including neurologic disorders such as stroke and epilepsy. The HIV pandemic increased the number of HIV centres in SSA where they have become pillars of the primary care system. In addition, the attitude to manage HIV can favour the management of other chronic conditions, both CDs and NCDs. Efforts to unify management of HIV and neurologic disorders at HIV centres are necessary. Task shifting, retention, and fees to access care are critical elements of the integration of HIV and NCD.

The task shifting process has to include nonmedical health workers; long-term programmes are needed. Western countries academic institutions and universities should rethink a *postcolonial* medical education.

Low retention in care reduces efficacy of interventions and the impact of funding in SSA and requires dedicated long-term strategies. Additionally, retention should be part of the analysis when evaluating cost-effectiveness of health interventions to chronic conditions in SSA.

More than half of the SSA population lives below the poverty threshold, at high risk of catastrophic economic expenditure when hit by diseases. Free access to ART is now recognized as a key measure to fight the HIV pandemic in SSA. A “pay-to-educate” approach (sometimes applied in SSA) needs to be carefully revised in light of ethical, health, long-term economic, and global effects. Disease-related impoverishment contributes to SSA population migration, and this affects neighbours and overseas countries.

In poor contexts, free of charge care can help link the patient to the health centre; as do personal relationships with health personnel. Patients and their families do not feel alone while facing illness and its consequences, and this can increase trust toward national institutions and governments, a way to feel part of the society and to build citizenship. It adds value to the unstable socioeconomic and political situation of some SSA countries.

The DREAM programme provides health services to prevent and treat HIV/AIDS and other chronic diseases such as stroke and epilepsy in SSA [58]. It offers continuous education and training to local health workers that involves European academic institutions and volunteer specialists. The good retention among DREAM patients is the result of multilevel strategies, including continuous education and task shifting. It is the precondition to deliver care for chronic conditions such as stroke and epilepsy.

Continuous education, task shifting, motivation, and networking help to prevent the attrition of DREAM health personnel.

Taken together, all of these strengthen the fragile health systems and contribute to the prevention of care disruption during COVID-19 and other pandemics in SSA.

## Figures and Tables

**Figure 1 ijerph-18-02766-f001:**
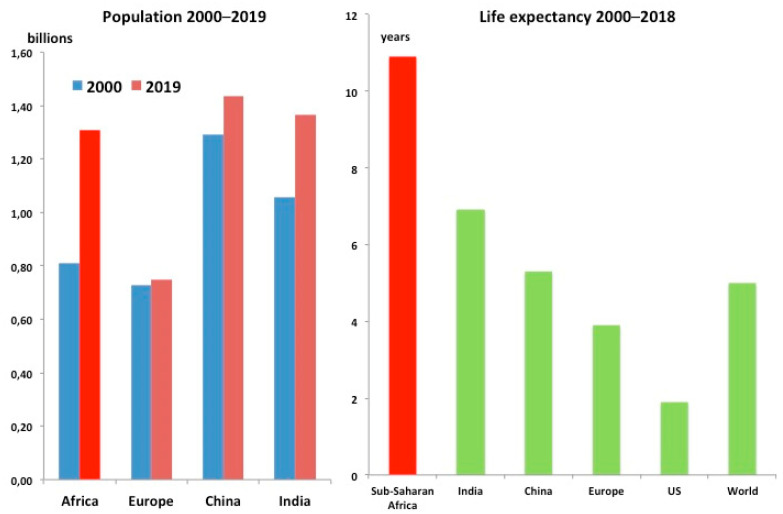
The epidemiologic transition: changes in the global population (left panel) and life expectancy increases from 2000 to 2018 (right panel) (ref. [1,2,3]).

**Figure 2 ijerph-18-02766-f002:**
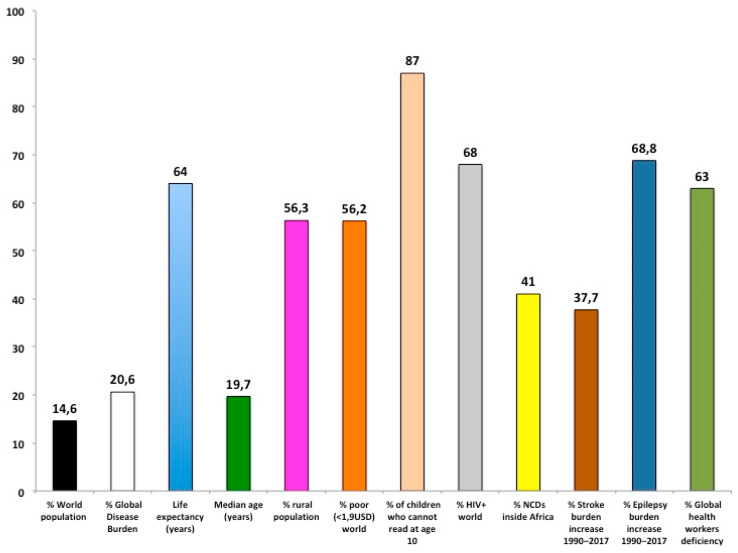
Overview of sub-Saharan Africa condition (see body text for details). NCDs: Noncommunicable diseases.

**Figure 3 ijerph-18-02766-f003:**
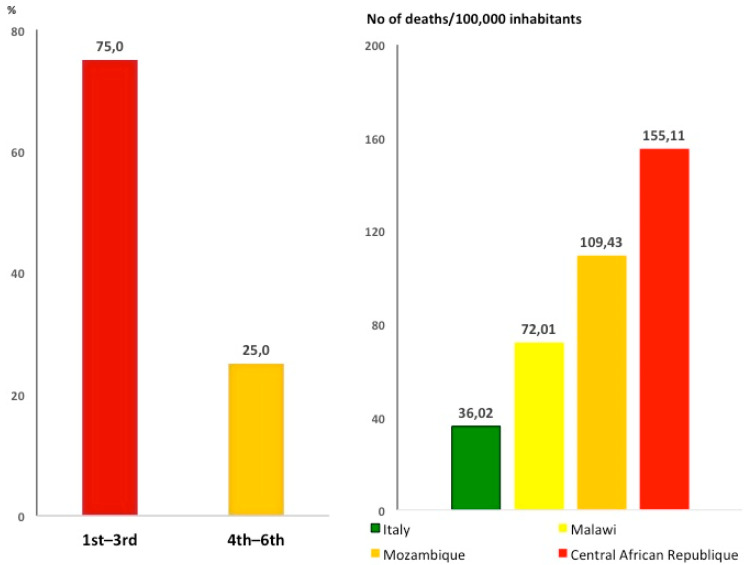
Stroke is a leading cause of death in sub-Saharan Africa. Left panel: stroke ranks as first to third leading cause of death in 75% of sub-Saharan countries (ref. [9]). Right panel: mortality for stroke in Italy compared to sub-Saharan Africa countries.

**Figure 4 ijerph-18-02766-f004:**
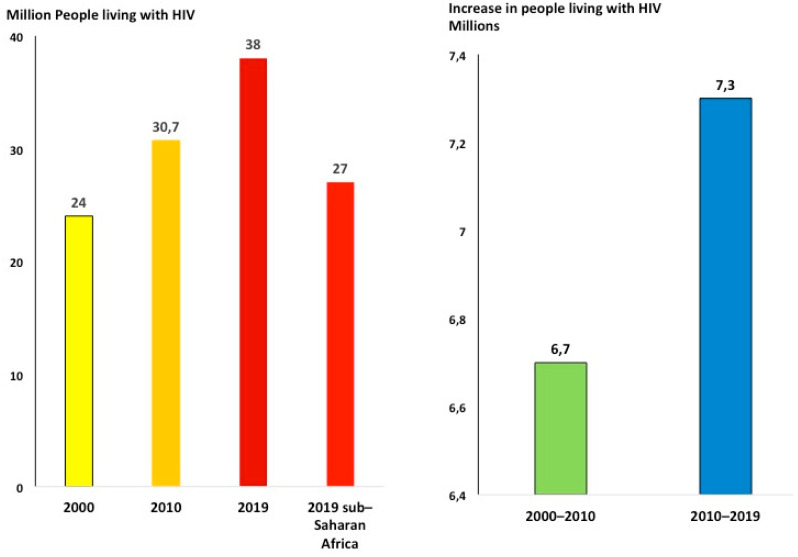
People living with HIV continue to increase (ref. [34]). More than two-thirds of people living with HIV across the world are in sub-Saharan Africa.

**Figure 5 ijerph-18-02766-f005:**
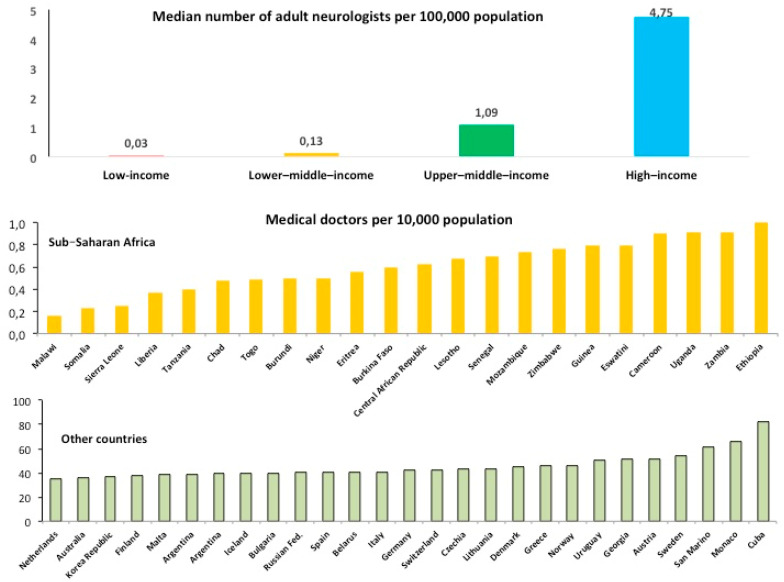
Health work force in sub-Saharan Africa. *Upper panel*. Shortage of neurologists in low-income countries (ref. [38]). In 23 African nations, there was one neurologist per 5 million population (From: Bower, J.H.; Zenebe, G. *Neurology*
**2005**, *64*, 412–415, doi:10.1212/01.WNL.0000150894.53961.E2). *Lower panel*. Number of medical doctors in sub-Saharan Africa compared to high income and other countries (http://apps.who.int/gho/data/node.main.HWFGRP_0020?lang=en, last accessed on 12 January 2021).

**Figure 6 ijerph-18-02766-f006:**
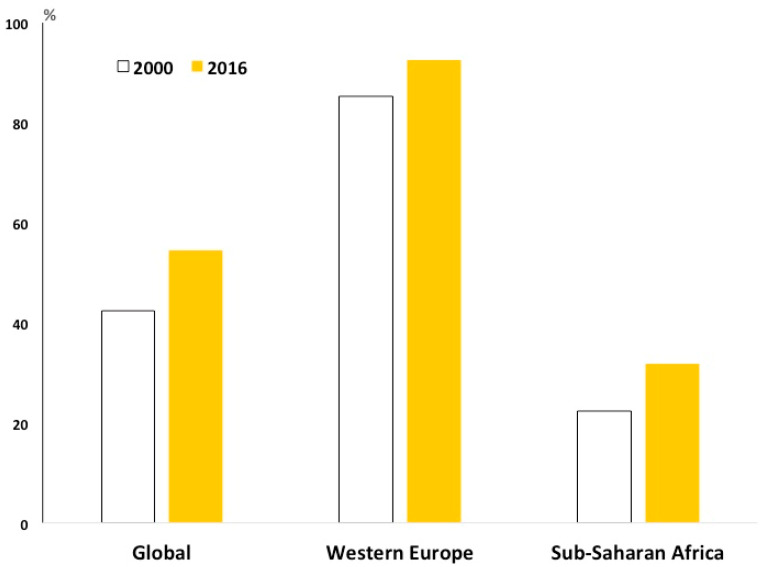
Healthcare access and quality (modified from ref. [43]).

**Figure 7 ijerph-18-02766-f007:**
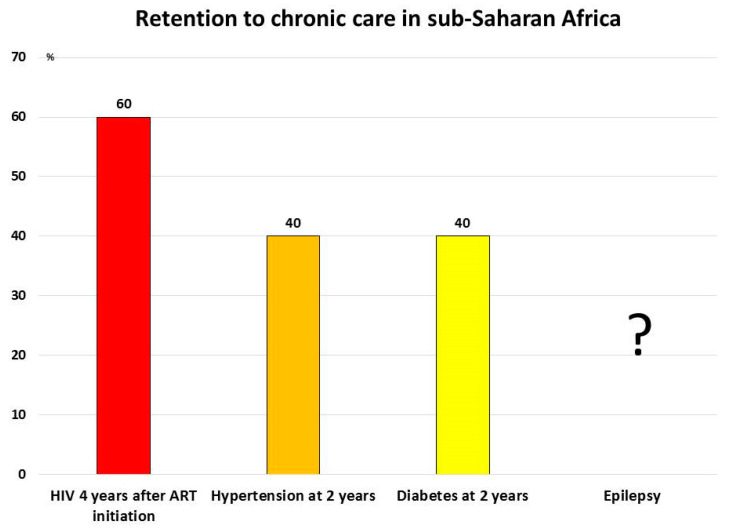
Percentage of patients under treatment after initiating chronic treatment in sub-Saharan Africa. The insufficient retention to chronic care hampers scaling up of treatment against HIV, hypertension, diabetes (ref. [50,51,52]). The insufficient retention deeply affects stroke prevention and epilepsy management.

**Figure 8 ijerph-18-02766-f008:**
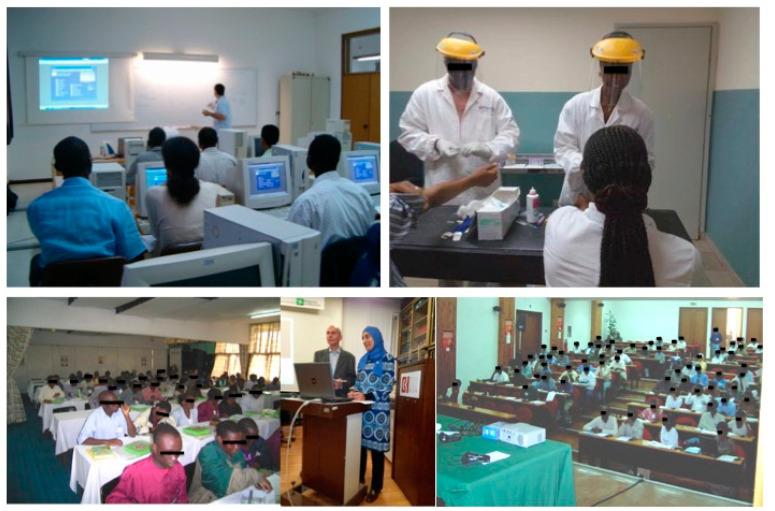
Continuous education is a pillar of the Disease Relief through Excellent and Advanced Means (DREAM) programme in 10 sub-Saharan African countries to provide health services to prevent and treat HIV/AIDS and other chronic diseases (ref. [58]). More than 10,000 African personnel have been trained by DREAM (ref. [58,59,60,61]).

**Figure 9 ijerph-18-02766-f009:**
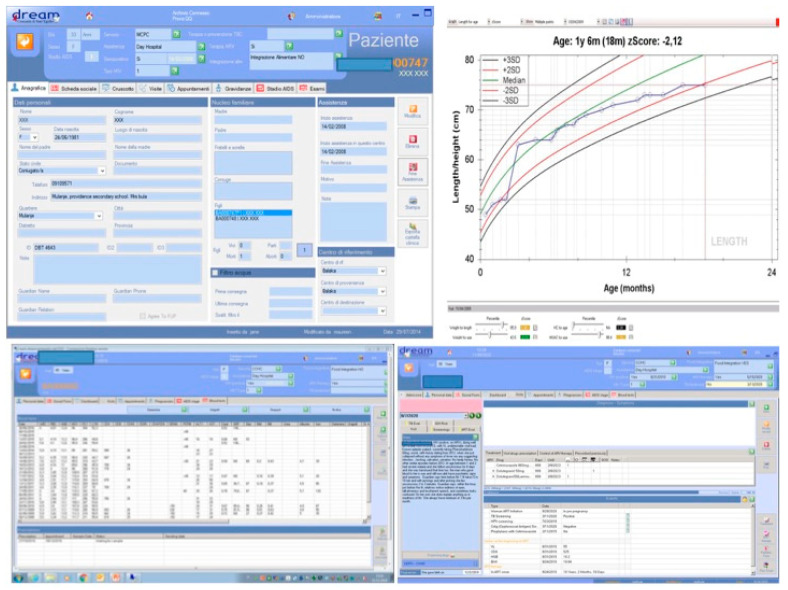
The DREAM software. Personal and social information, visits, investigations, examination, laboratory data, adherence to the antiretroviral treatment, etc. are stored in the computerized clinical record of the patient.

**Figure 10 ijerph-18-02766-f010:**
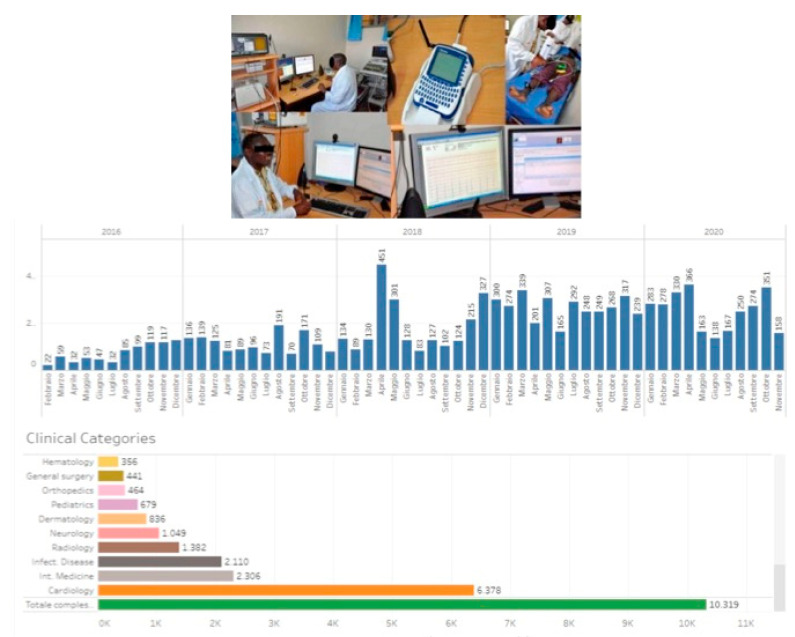
A telemedicine system with 30 workstations in sub-Saharan Africa assists clinical activities at DREAM to increase quality of health care, including remote areas (ref. [60,65]).

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
