# Peer review of "Pandemics and Burden of Stroke and Epilepsy in Sub-Saharan Africa: Experience from a Longstanding Health Programme"

_ijerph, 2021, doi:10.3390/ijerph18052766_

Round 1

Reviewer 1 Report

General comments

The manuscript addressed an important topic specifically in the context of HIV. The title gives the impression that neurological disorders to be covered will be very comprehensive, instead the paper focuses mainly on stroke and epilepsy. Therefore, the title should be revised accordingly. I would suggest: Burden of stroke and epilepsy in sub-Sahara Africa: experience from a longstanding health program. The authors shared lot of information but the paper should focus on stroke and epilepsy in sub-Saharan Africa and within the context of HIV. The paper will be more streamlined and the author can focus on how their program manage stroke and epilepsy in the context of HIV. The paper should not focus or elaborate on COVID. The authors could mention how COVID has affected their services or how they have managed the two major neurological disorders during COVID time. I think the case of Malawi and COVID dilutes the focus of the paper as well as any focus on NCD although important. The manuscript definitely needs more editing (see detailed inputs below).

Detailed inputs

Line 40: environmental determinants also are incriminated in changes of health disease

Line 89: 80% of epileptic patients

Line 90: represent

Line 91:  with epilepsy is three times higher than in the general population and even greater in SSA

Line 105: may undermine

Line 106: which lead parents of epilepsy children to see the services of traditional healers

Line 109:  very often suffer from stigma

Line 110: resources to treat an epileptic patient

Line 111: This disparity

Line 117: permanently

Line 121:  the result of longer life expectancy due to more effective antiretroviral treatment regimens and the occurrence of new infections

Line 124: the number of PLWH

Line 125: what has been

Line 131:  To reduce

Line 133: All patients living with HIV in SSA

Line 138: will be

Line 144 & 145: This sentence needs to be edited for clarity

Line 148:  the number of medical doctors is limited and

Line 210: cannot afford the cost of health care

Line 211 – replace so that by hence

Line 236: shared field work and practice

Line 236 - 237: including distancing learning, continuing education and clinical cases discussions.

Line 257:   counseling instead of advices

Line 259: is provided – Regarding cervical cancer, thousands of women….. for early diagnosis and treatment,

Line 261 – 263: also with ---- etc. Please, edit as necessary to make this sentence more comprehensible

Line 273: life events …………. And food shortage which can affect adherence to HIV treatment

Line 274 - 278: can be shortened: they provide peer counseling before, during and after initiation of treatment. They also assist in patient care (recording of vital signs and weight) and follow-up

Line 291: strategies that reduce or have impact on neurologic diseases occurrence should be described since the sub heading refer to strategies for HIV+ young patients.

Line 293: 6.2.3 this is not very relevant to the main topic – no relation with neurologic disorder described

Figure 10 A – a high resolution picture will be needed for the picture of the clinical categories and repartition by month

Line 349:  People 60 years of age and older

Line 351: clarify results 5 to 6 times or 5.6 times

Line 361: use like instead of as – many of the COVID-19 patients who died

Line 403: check if 10,81 USD or 1,081 USD?

Line 404: remove the from the ART

Line 409: not clear what all of these refer to: ART? Test and treat? Both?

Line 411: The prevalence of epilepsy in the country is very high, around 2.8% and access to anti-epileptic treatment and care is limited.

Line 420: is sharing his knowledge in Neurology at country level

Line 443: remotely instead of from remote

Line 445: the HIV pandemic increased the number of ….

Line 459: Task shifting

Line 458: these challenges

Line 460: are critical elements of the integration of HIV and NCD

Author Response

Reviewer 1

Comments and Suggestions for Authors

General comments

The manuscript addressed an important topic specifically in the context of HIV.

We wish to thank the referee for her/his positive and encouraging comments

  1. The title gives the impression that neurological disorders to be covered will be very comprehensive, instead the paper focuses mainly on stroke and epilepsy. Therefore, the title should be revised accordingly. I would suggest: Burden of stroke and epilepsy in sub-Sahara Africa: experience from a longstanding health program.

Reply:

We follow the suggestion of the reviewer, the title has now been changed as follows: Pandemics and burden of stroke and epilepsy in sub-Saharan Africa. Experience from a longstanding health program.

We leave the term “pandemic” because:

  • the manuscript reports on to how HIV pandemic increases epilepsy and stroke burden in sub-Saharan Africa, and
  • how COVID further increases their burden through care disruption - a feared effect to the large and fragile HIV population.
  1. The authors shared lot of information but the paper should focus on stroke and epilepsy in sub-Saharan Africa and within the context of HIV.

Reply:

We thank the referee for appreciating the information we give in our manuscript.

Maybe some arguments in the manuscript appear to be not strictly focused on epilepsy and stroke in sub-Saharan Africa. We now better clarify why some arguments are worth to be reported and can be useful to neurologists dealing in sub-Saharan Africa. There, the vast majority of stroke and epilepsy are managed at primary care level. Section 5 describes main barriers in developing primary care programs for chronic conditions that include epilepsy and stroke. In the subsequent section 6 we then report hints from a longstanding HIV program, DREAM, in sub-Saharan Africa that can help to manage epilepsy and stroke in HIV+ patients. In practice malnourished epileptic HIV+ children in poor families - not so unusual in those contexts -  will benefit of HIV management and good neurologic diagnosis and appropriate pharmacological treatment (delivered by educated and trained local health workers); but those children – and their families - will also need counselling to adherence and retention; nutrition support can be useful. All of those means a personalized care strategy detailed in section 6.2.      Another example: preventing stroke in HIV+ hypertensive patients may require neurologist’ and cardiologist’ advices from remote; integrated telemedicine, ie teleneurology and telecardiology is a powerful tool to improve patients’ condition but need well trained local health workers to convey specialists’ advices to patients. To establish good communication between neurologists and local health providers is necessary (reference 50)

Inadequate patients’ retention is a major sources of failure in sub-Saharan Africa (Kones R et al. Lancet 2019; 393: 105-6) so we feel that  chapters 5.2 and 6.2 dealing with retention are necessary.

European neurologists approaching stroke and epilepsy in HIV+ sub-Saharan Africa patients should become familiar with those issues; reporting effective solutions can help preventing frustrations and abandon of European specialists to leave problems as they are.

  1. The paper will be more streamlined and the author can focus on how their program manage stroke and epilepsy in the context of HIV.

Reply:

We thank the referee for the comment. To better explain how the DREAM program manages stroke in the context of HIV we have now added a dedicate section 6.4 “Stroke prevention at DREAM” as follows: “Arterial blood pressure (BP), blood glucose, hemogram, creatinine, urea, urinanalysis and body mass index (BMI) are regularly checked in all patients. All patients receive counselling for life-style change. In patients harbouring cardiovascular risk factors as hypertension, diabetes, kidney failure etc. electrocardiogram (ECG) is regularly performed and ECGs are evaluated by remote cardiologists employing the dedicated telemedicine platform. Controls and visits are tailored according to risk factors and clinical condition including BP, BMI, viral load, adherence, blood glucose, cholesterol, smoke and drinking habits, social condition.” Page 12, page bottom.

In general the stroke and epilepsy programs at DREAM benefits form three common pillars reported in section 6: a. education and training of local health workers,  b. programs to enhance patients’ retention (see also reply to question 2; Lancet 2019; 393: 105-6) and c. telemedicine.

In section 7 “Disruption of care under COVID-19 pandemic in sub-Saharan Africa affect stroke and epilepsy” we report another aspect of stroke and epilepsy management in the context of HIV in the DREAM program, that is granting patients high quality drug supply without stock interruption during the pandemic,  a serious issue in many African regions. We can add that DREAM patients appreciate the efforts to grant them high quality drugs without interruption, and this increases their trust and retention to the program.   The link between trust and retention there is important. This is an interesting topic, a lot to do there - we do not want to make the story long.

  1. The paper should not focus or elaborate on COVID.

Reply:

We would leave the description about COVID induced care disruption affects stroke and epilepsy burden in the HIV+ patients: this is a day-by-day increasing problem there. In this regard (care disruption provoked by pandemics) Ebola taught us something (ref. 48, 49, 67 and 68). Hopefully to uncover that might contribute to improve actions in due time.

  1. The authors could mention how COVID has affected their services or how they have managed the two major neurological disorders during COVID time.

Reply:

We thank the referee for this important question. 

In section 8, on pag 14, line 391-399 we reported that “a number of activities including medical triage by telephone, individualised counselling, differential programs according to clinical and social conditions, home-based care for patients with advanced HIV, longer refills and/or home delivery of chronic medication for stable patients, focused training to health workers, enhanced link with local communities, stigma reduction messages, special attention to prevent stock-outs, accelerated integration of CDs and NCDs diseases”.  

In addition we have better clarified as follows “patients harboring cardiovascular risk factors as hypertension, diabetes, arrhythmias, obesity etc receive personalized appointments for check and visits according to their condition. Epileptic patients monitoring is tailored to their pre-existing adherence and responsiveness to antiepileptic drugs. Age, general and social condition, distance from the health centre are also considered” (page 14, from the last the 3 lines at the bottom of the page to page 15, the fist 2 lines at the top).

  1. I think the case of Malawi and COVID dilutes the focus of the paper as well as any focus on NCD although important.

The case of Malawi can be seen as a real life example, a model to treat chronic disorders at country level in poor resources countries. Along the years HIV chronic care requested integrated programs to manage HIV+ patients developing chronic diseases as epilepsy and stroke. United Nations indicated to take care of those patients at the HIV centre. The DREAM program is actively preventing COVID related care disruption on stroke and epilepsy - a resilient primary care system.

  1. The manuscript definitely needs more editing (see detailed inputs below).

Detailed inputs

Line 40: environmental determinants also are incriminated in changes of health disease

We have modified the text accordingly

Line 89: 80% of epileptic patients

We have modified the text accordingly

Line 90: represent

We have modified the text accordingly

Line 91:  with epilepsy is three times higher than in the general population and even greater in SSA

We have modified the text accordingly

Line 105: may undermine

We have modified the text accordingly

Line 106: which lead parents of epilepsy children to see the services of traditional healers

We have modified the text accordingly

Line 109:  very often suffer from stigma

We have modified the text accordingly

Line 110: resources to treat an epileptic patient

We have modified the text accordingly

Line 111: This disparity

We have modified the text accordingly

Line 117: permanently

We have modified the text accordingly

Line 121:  the result of longer life expectancy due to more effective antiretroviral treatment regimens and the occurrence of new infections

We have modified the text accordingly

Line 124: the number of PLWH

We have modified the text accordingly

Line 125: what has been

We have modified the text accordingly

Line 131:  To reduce

We have modified the text accordingly

Line 133: All patients living with HIV in SSA

We have modified the text accordingly

Line 138: will be

We have modified the text accordingly

Line 144 & 145: This sentence needs to be edited for clarity

We have modified the text accordingly

Line 148:  the number of medical doctors is limited and

We have modified the text accordingly

Line 210: cannot afford the cost of health care

We have modified the text accordingly

Line 211 – replace so that by hence

We have modified the text accordingly

Line 236: shared field work and practice

We have modified the text accordingly

Line 236 - 237: including distancing learning, continuing education and clinical cases discussions.

We have modified the text accordingly

Line 257:   counseling instead of advices

We have modified the text accordingly

Line 259: is provided – Regarding cervical cancer, thousands of women….. for early diagnosis and treatment,

We have modified the text accordingly

Line 261 – 263: also with ---- etc. Please, edit as necessary to make this sentence more comprehensible

We have changes as follows “Local and national campaigns on TV, newspapers and social media have been employed; governments, district hospitals, and health facilities have been involved in those campaigns”

Line 273: life events …………. And food shortage which can affect adherence to HIV treatment

We have modified the text accordingly

Line 274 - 278: can be shortened: they provide peer counseling before, during and after initiation of treatment. They also assist in patient care (recording of vital signs and weight) and follow-up

 We have modified the text accordingly: “The expert clients provide peer counseling before, during and after initiation of treatment. They also assist in patient care (recording of vital signs and weight) and follow-up and”

Line 291: strategies that reduce or have impact on neurologic diseases occurrence should be described since the sub heading refer to strategies for HIV+ young patients.

We have now reported as follows: “to help with school activities and to talk each other: this helps to win isolation and stigma. Some of them give talks to schools, disclose their condition to show that a good life is possible with HIV. This strenghtens their motivation to maintain good adherence so that constant undetectabe viral load much reduces the risk to develop neurologic disorders.” (page 10, bottom)

Line 293: 6.2.3 this is not very relevant to the main topic – no relation with neurologic disorder described

With respect: it is not easy to treat diseases in malnourished patients; many need to walk hours to reach the health centre, many are women with children. We also have a nutrition program studied by European specialists to teach patients how to use and cook the local food.

Figure 10 A – a high resolution picture will be needed for the picture of the clinical categories and repartition by month

We have tried in several way (it is a screenshot) but unfortunately this is the best quality, sorry

Line 349:  People 60 years of age and older

We have modified the text accordingly

Line 351: clarify results 5 to 6 times or 5.6 times        

Thank you, it is 5.6, the text has been edited

Line 361: use like instead of as – many of the COVID-19 patients who died

We have modified the text accordingly

Line 403: check if 10,81 USD or 1,081 USD?

We have checked, it is 10.81USD, thank you

Line 404: remove the from the ART

Removed

Line 409: not clear what all of these refer to: ART? Test and treat? Both?

We now better clarify : “… nowadays 79% of the HIV+ patients receive ART [85]. These patients are at increased risk to develop stroke..”

Line 411: The prevalence of epilepsy in the country is very high, around 2.8% and access to anti-epileptic treatment and care is limited.

We have modified the text accordingly

Line 420: is sharing his knowledge in Neurology at country level

We have modified the text accordingly

Line 443: remotely instead of from remote

We have modified the text accordingly

Line 445 (maybe it is 455): the HIV pandemic increased the number of ….

We have modified the text accordingly

Line 459: Task shifting

We have modified the text accordingly

Line 458: these challenges

We have modified the text accordingly

Line 460: are critical elements of the integration of HIV and NCD

We have modified the text accordingly

Reviewer 2 Report

Very interesting paper. I've few suggestions/questions: - Figure 1: I don't understand the right-side graph: is it life expectancy or the increase in LE between 2000 and 018? - Paragraph 2: you assert that HIV is the second main risk factor for stroke based on reference n°9. but I've not be able to find this information following the link. So, can you give reliable reference to confirm yous statement? - Paragraph 6.1 How about HPV vaccination in SSA countries? Have you some informattion about that? - 'Conclusion: Postcolonial' medical education is not politically correct, in my opinion. Can you delete?

Author Response

Reviewer 2

Comments and Suggestions for Authors

Very interesting paper.

We wish to thank the referee for her/his positive and encouraging comments on our manuscript.

I've few suggestions/questions:

  1. Figure 1: I don't understand the right-side graph: is it life expectancy or the increase in LE between 2000 and 018?

Reply:

We have now modified the legend to figure 1 as follows: “The epidemiologic transition. Changes in global population and life expectancy increase from 2000 to 2018(ref. 1-3)”.

  1. Paragraph 2: you assert that HIV is the second main risk factor for stroke based on reference n°9. but I've not be able to find this information following the link. So, can you give reliable reference to confirm yous statement?

Reply:

We thank the referee for the observation, the right reference is n. 10 (Benjamin et al. Neurology 2016) and this is now corrected in the body text. Sorry the mistake.

  1. Paragraph 6.1 How about HPV vaccination in SSA countries? Have you some informattion about that?

Reply:

We thank the referee for the substantial observation. Cervical cancer is the leading cause of deaths attributed to tumors in Africa. Even if the HPV vaccination has  been evaluated to be very cost/effective in low income countries as in sub-Saharan Africa, there is still a lot to do. Denny et al. (Lancet 2017; 389: 861–70) report:Several model-based analyses have been done to assess the health and economic effects of HPV vaccination in LMICs. One indicated that 690 000 incident cervical cancers and 420 000 cervical cancer deaths would be prevented during the lifetime of a cohort of 58 million girls who received HPV vaccination at age 12 years in 179 countries, most of which would be LMICs. The cost of such vaccination was estimated to be $4 billion.43 Goldie and colleagues44 also concluded that HPV vaccination is likely to be very cost-effective in the poorest countries of the world.

  1. 'Conclusion: Postcolonial' medical education is not politically correct, in my opinion. Can you delete?

 Reply:

We understand this comment and agree the wish to not disturbing readers. On the other side we try reassure the referee. The term “postcolonial” is commonly used without the aim to offend anyone; the oxford dictionary definition of the term “postcolonialism (postcoloniality, postcolonial theory)” is:  

The study of the cultures of countries and regions, especially in Africa, Asia, and Latin America, whose histories are marked by colonialism, anti-colonial movements, and the transition to independence during the 20th century, and the study of their present-day influence on the societies and cultures of former colonizers …https://www.oxfordreference.com/view/10.1093/acref/9780199587261.001.0001/acref-9780199587261-e-0543

and the Cambridge Dictionary definition is: “from or relating to the period after colonialism” https://dictionary.cambridge.org/dictionary/english/postcolonial

In case the referee is still not convinced by our explanation we accept the Editor to erase the term postcolonial.

Reviewer 3 Report

Pandemics and burden of neurological disorders in Sub-Saharan Africa. Experience from a longstanding health program

Decision: Rejected

Abstract: There are major concerns in the abstract. For example, the authors should use a numerical form, e.g. 80%, to describe eighty%. Furthermore, the abstract also should not be started with a statistical number. Another major issue of this article is that the authors failed to clarify the objectives of this study. Sentences are not linked in the abstract and failed to convey a compelling message. Besides, top of the first page, the author claimed the article as a "Review"; however, it is presented as commentary. There is misconnection between article's claim and the way it was written. The author failed to show the methods used to collect data of this review article/commentary.

The epidemiologic transition: Overall, the manuscript is presented with overwhelming graphs. In figure 1, there is a spelling mistake of China. It was written as “Cina”. The authors also did not mention which country from Sub-Saharan Africa they are focusing on and why. Most of the graphs do not have a labeled X-axis.

Epilepsy:  Although epilepsy is an issue in Sub-Saharan Africa, however, it is not clear the relevance of epilepsy with this topic. The authors failed to make compiling argument that encourages readers to read this article.

The double burden of HIV: Another major concern with this manuscript is that it does not state how HIV impact Africa. The article lacks statistics on HIV in Africa and the connection between epilepsy, stroke, and HIV.

Unifying Care of HIV: In line 144, the author left grammatical correction which is not admissible. In figure 5, there is a question mark, which shows a lack of proofreading before submission. This manuscript is poorly written and full of grammatical errors and some spelling mistakes.

The Dream program: The authors failed to state if they obtained consent from people to publish their pictures. However, what is the relevance of these pictures in the manuscript? Furthermore, some parts of the article are too short, and some are too long, and overall it lacks consistency.

Conclusion: A conclusion should be short, concisely written, and it should address the objectives of the articles. The article does not follow a logical process.  Are authors looking for COVID-19 impact on HIV and other neurological care access? This manuscript also lacks statistical data on the effects of Telemedicine on the burden on the neurological disorder.

Since the authors did not adequately mention the purpose of this study in the abstract and the content, the conclusion cannot be fully supported. . Moreover, the manuscript is full of grammatical errors and lacks proper labeling of the graphs. Overall, the authors heavily included background information. However, the article failed to show any new things added to the literature.

Author Response

.please see attachment

Round 2

Reviewer 1 Report

The authors have addressed my comments. However I still feel that the section nutrition is not adding much to the main topic of the manuscript. The manuscript will benefit from additional editing and proofreading.

Author Response

Reviewer 1

We are much grateful to Referee 1 for the efforts to improve our manuscript.

Here is our point-by-point reply and changes done to the manuscript.

Line 21: Eighty percent.

Reply: line 21: “percent” has been inserted and “%” has been deleted as suggested.

Line 26: weak instead of insufficient – Many standalone HIV health centres (and remove outside hospitals).

R: line 26: “weak” has been inserted instead of “insufficient” and “Many standalone HIV health centres” and “outside hospitals” has been removed as suggested.

Line 28: is overwhelming SSA fragile health systems

R: Line 28: “overwhelming” has been inserted and “hitting” has been removed as suggested.

Line 33: management of NCDs including -------. We described challenges and solutions to address disruption and excess deaths from these diseases during the ongoing COVID-19 pandemic.

R: Line 33: “of NCDs including” has been inserted and “of NCD to include” deleted. Line 33-35: “We described challenges and solutions to address disruption and excess deaths from these diseases during the ongoing COVID-19 pandemic” has been inserted and “Main issues of the process are presented also in the light of the experiences to limit care disruption and excess deaths under COVID-19 pandemic” has been deleted as suggested

Line 42: in changes of disease burden

R: Line 42: “disease burden.” has been inseretd and “health disease” has been deleted.

Line 48: increased the burden of non-communicable diseases that…

R: Line 48: “increased the burden…” has been inserted and “contributed to increase the burden” has been been deleted.

Figure 2: title need to be more precise: Overview of what? Demographic change and burden of NCD and neurologic disorders in SSA? Consider removing the title above the graphs – it is repeated below for all graphs.

R: The title on graph 2 has now been removed.

Titles above the graphs have been removed from Figure 1, 2, 3, 4, 6, 8, 9, 10.

Line 60: greatly increased

R: Line 60: “greatly” has been inserted “much” has been deleted.

Line 64: death rate by stroke was 37 per 100,000 inhabitants in SSA

R: Line 64: “death rate by stroke was” has been inserted and “stroke provoked” has been deleted.

Line 66: is among the three leading causes of deaths

R: Line 66: “is among the three leading…” has been inserted and “first 3” has been deleted.

Line 73: below 45 years of age

R: Line 73: “of age” has been added.

Line 81: affects between 16% and 40%

R: Line 81: “affects between 16% and 40%” has been inserted and “from … to..” deleted.

Line 93: the risk of premature death among people with

R: Line 93: “among people” has now been added.

Line 126: decreased instead of reduced

R: Line 126: “decreased” has been inserted and ”reduced” deleted

Line 213: after initiating treatment

R: Line 213: “initiating” has been inserted and “initiation of” deleted

Line 292: depression and anger towards

R: Line 292: “anger” has been inserted and “rage” deleted

Line 298: to talk to each other … This helps fight isolation and stigma

R: Line 298: “to” has been inserted; “fight” has been inserted and “to win” deleted.

Line 299: some of them? Who? DREAM healthcare workers? Expert clients? HIV+ adolescents?

R: Line 299: “HIV+ adolescents” has been inserted and “them” deleted.

Line 302: I really think that the malnutrition section is not adding much to the paper. I understand that it is important but the paper needs to have an area of focus. This section is not doing justice to the whole issue of nutrition, infant neurological development and HIV. Removing this section will not diminish the quality or focus of the paper.

R: We agree with the referee that the section on malnutrution not give justice to the whole issue of nutrition, infant neurological development and HIV: the purpose is not to give an indepth analysis of all those aspects. We think useful to leave this section because information on nutrition and diseases as stroke, epilepsy and HIV are usually presented as separate issues while in sub-Saharan Africa those are not rarely encountered at the same time in the same patient.

To divide things may well not represent real world at HIV centres there.

As western specialists we are used (and asked) to take in consideration only one aspect of the problem, that is good for many aspects. But for other aspects this may weaken our ability to unify thoughts, ideas and then proposals.

We hope to help readers to have an olistic view of patients (and problems) in sub-Saharan Africa.

Line 364: health systems

R: Line 364: the article “the” has been deleted

Line 370: 25.3% and 4.7% of COVID patients or general population?

R: Line 370: “general” has now been added before “population”

Line 372: ..… in SSA was 5.6 times higher

R: Line 372: “was” has been inserted and “results” deleted

Line385: COVID 19 spread

R: Line385: “spread” has been added before COVID-19

Line 387: counterfeit medicines

R: Line 387: “counterfeit” has been inserted and “falsified” deleted

Line 399: in resource poor countries

R: Line 399: “in” has been inserted and “to for” deleted.

Line 440: European Neurologist volunteer

R: Line 440: “European Neurologist volunteer” has been inserted “and “European voluntary neurologists” deleted.

Line 445: consultations

R: Line 445: “consultations” has been inserted

Line 466: COVID genome research? Is it a sequencing of COVID 19 or is it a PCR diagnostic test?

R: Line 466: it is now clarified it is a PCR diagnostic test

Line 479: as well as local disease patterns

R: Line 479: “as well as local disease patterns” has been inserted and “and their diseases pattern is being modified accordingly” deleted

Line 486: consider removing “these….global health”.

R: Line 486: “these….global health” has been removed

Line 489: the Task shifting process

R: Line 489: “shifting” has been inserted and “shift” deleted

Line 493: should be part of the analysis

R: Line 493: “the” has been inserted

Line 512: task shifting

R: Line 512: “shifting” has been inserted and “shift” deleted

Line 514: task shifting

R: Line 514 “shifting” has been inserted and “shift” deleted

Reviewer 3 Report

 it can be accept

Author Response

Thank you